# The relationship between a plant-based diet and mental health: Evidence from a cross-sectional multicentric community trial (LIPOKAP study)

**Fahimeh Haghighatdoost**[1‡], **Atena Mahdavi**[2‡], **Noushin Mohammadifard**[3*],
**Razieh Hassannejad**[4], **Farid Najafi**[5], **Hossein Farshidi**[6], **Masoud Lotfizadeh**[7],
**Tooba Kazemi**[8], **Simin Karimi**[9], **Hamidreza Roohafza**[10], **Erika Aparecida Silveira**[11,12],
**Nizal Sarrafzadegan**[1], **Cesar de Oliveira**[11]

1 Isfahan Cardiovascular Research Center, Cardiovascular Research Institute, Isfahan University of Medical Sciences, Isfahan, Iran, 2 Food Security Research Center and Department of Community Nutrition, School of Nutrition and Food Science, Isfahan University of Medical Sciences, Isfahan, Iran, 3 Interventional Cardiology Research Center, Cardiovascular Research Institute, Isfahan University of Medical Sciences, Isfahan, Iran, 4 Hypertension Research Center, Cardiovascular Research Institute, Isfahan University of Medical Sciences, Isfahan, Iran, 5 Research Center for Environmental determinants of Health, Health Institute, Kermanshah University of Medical Sciences, Kermanshah, Iran, 6 Hormozgan Cardiovascular Research Center, Hormozgan University of Medical Sciences, Bandarabbas, Iran, 7 Social Determinants of Health Research Center, Shahrekord University of Medical Sciences, Shahrekord, Iran, 8 Cardiovascular Diseases Research Center, Birjand University of Medical Sciences, Birjand, Iran, 9 Heart Failure Research Center, Cardiovascular Research Institute, Isfahan University of Medical Sciences, Isfahan, Iran, 10 Cardiac Rehabilitation Research Center, Cardiovascular Research Institute, Isfahan University of Medical Sciences, Isfahan, Iran, 11 Department of Epidemiology & Public Health, Institute of Epidemiology & Health Care, University College London, London, United Kingdom, 12 Postgraduate Program in Health Sciences, Faculty of Medicine, Federal University of Goiás, Goiânia, Brazil

‡ HF and MA contributed equally and are both first author on this work.
* nmohammadifard@gmail.com

## Abstract

### Background

Dietary patterns emphasizing plant foods might be neuroprotective and exert health benefits on mental health. However, there is a paucity of evidence on the association between a plant-based dietary index and mental health measures.

### Objective

This study sought to examine the association between plant-based dietary indices, depression and anxiety in a large multicentric sample of Iranian adults.

### Methods

This cross-sectional study was performed in a sample of 2,033 participants. A validated food frequency questionnaire was used to evaluate dietary intakes of participants. Three versions of PDI including an overall PDI, a healthy PDI (hPDI), and an unhealthy PDI (uPDI)

**Data Availability Statement:** Data are available from the Isfahan Cardiovascular Research Institute,

Institutional Data Access / Ethics Committee (contact via icri.researchers@gmail.com) for researchers who meet the criteria for access to confidential data.

**Funding:** This study received funding from Pfizer company (#11531879) and Dr. Cesar de Oliveira, supported by the Economic and Social Research Council (ESRC) (#ES/T008822/11). The funders were involved in the study design, collection, analysis, interpretation of data, the writing of this article or the decision to submit it for publication.

**Competing interests:** The authors have declared that no competing interests exist.

were created. The presence of anxiety and depression was examined via a validated Iranian version of the Hospital Anxiety and Depression Scale (HADS).

## Results

PDI and hPDI were not associated to depression and anxiety after adjustment for potential covariates (age, sex, energy, marital status, physical activity level and smoking). However, in the crude model, the highest consumption of uPDI approximately doubled the risk of depression (OR= 2.07, 95% CI: 1.49, 2.87; P<0.0001) and increased the risk of anxiety by almost 50% (OR= 1.56, 95% CI: 1.14, 2.14; P= 0.001). Adjustment for potential confounders just slightly changed the associations (OR for depression in the fourth quartile= 1.96; 95% CI: 1.34, 2.85, and OR for anxiety in the fourth quartile= 1.53; 95% CI: 1.07, 2.19).

## Conclusions

An unhealthy plant-based dietary index is associated with a higher risk of depression and anxiety, while plant-based dietary index and healthy plant-based dietary index were not associated to depression and anxiety.

## Introduction

Psychological disorders, including depression and anxiety, are one of the main public health issues globally due to their high prevalence and poor outcomes [1]. Depression is a major cause of disease and disability affecting over 300 million people around the world [2]. In addition, the percentage of people suffering from mood disorders or anxiety is just over 15% [3]. It is well established that psychological disorders adversely influence an individual's health, quality of life, lifespan, and alter dietary behaviors [4].

Emerging evidence suggests that diet plays a crucial role in mental health status [5]. Amongst various dietary patterns, the health benefits of a vegetarian diet have been widely indicated [6]. However, plant-based dietary patterns, depending on their definition (excluding any type of animal products or just relying on high factor loadings for plant-based foods) may vary, resulting in conflicting findings [7]. For instance, a recent meta-analysis suggested that while vegetarian and semi-vegetarian diets may increase the risk of depression, lacto-ovo vegetarian was not significantly related to depression.[8]

Similarly, plant-based diet indices (PDIs), calculated using a scoring system, may also be associated with better health status [9–12], but it is of note that all plant-derived foods are not necessarily healthful (e.g. refined grains and potatoes) and possibly increase the risk of various diseases. Therefore, three diverse versions of PDI, taking into account the health effects of plant foods, namely, an overall PDI, a healthy plant-based diet index (hPDI), and an unhealthy plant-based diet index (uPDI) were developed [13]. It has been indicated that hPDI and uPDI are differently related to the risk of various diseases. Particularly, two small studies among healthy Iranians [14] and diabetic Iranians [15] suggested an inverse association for hPDI but a direct association for uPDI with depression and anxiety [14].

To date, only few studies have examined the associations between plant-based diets and mental disorders [14, 16–18]. These studies have been mainly conducted in a small sample size and a specific population which result in overestimation of findings (e.g. in diabetic patients) and restricted generalizability. Moreover, vegetarian and vegan diets are not popular in

Iranians. Due to limited number of studies examining the association of plant-based diets with mental disorders, as well as differences in approaches to define plat-based diets, the associations may differ from one population to another. In addition, dietary intakes, like other lifestyle factors, vary from one population to another and therefore it is possible that such differences affect the associations. In particular, Iranians' diet is mainly based on carbohydrate (above 60% of daily energy intake), in particular refined and simple sugar, while fats and proteins have moderate contribution. As a consequence, studies examining the role of carbohydrate and their sources beside fats and protein would be relevant. Moreover, exploring the benefits of plant-based diets in a general population can provide an insight into some beneficial approaches to ameliorate health status and reduce diseases burden. Therefore, in the current study, we aimed at investigating the association between PDI, depression and anxiety, and examining how the associations differed for hPDI and uPDI in a multicentric sample of Iranians.

## Methods

### Study population

This study was conducted in the framework of the Knowledge and Practice of Dyslipidemia Prevention, Management, and Control study (LIPOKAP) [19]. The LIPOKAP project is a community-based trial with the primary aim of improving participants' knowledge and practice in terms of dyslipidemia. The current analysis was performed on the baseline data of the LIPOKAP project as a cross-sectional study.

Sample size was calculated using the following formula considering the error of type I=5%, and study power=90%. Standard error for participants' knowledge regarding dyslipidemia was calculated using the variation range (100-0)/6= 16.7 and d (absolute margin of error) was considered to be= 1.2%,

$$n = \frac{\left(z_{1-\frac{\alpha}{2}} + z_{1-\beta}\right)^2 s^2}{d^2}$$

Estimated sample size was equal to 2000 subjects.

Participants were selected from five cities in Iran (Isfahan, Birjand, Bandar abas, Kermanshah, and Shahrekord) from February 2018 to July 2019. The sample size was estimated according to a stratified multistage random cluster sampling method. The adequate sample size was first calculated and then doubled due to having different clusters. The final sample size for each city was determined based on population distribution in different cities and in the urban and rural areas of each city. From among available clusters in health care centers, clusters were selected on a random basis. Finally, according to the distribution of population across different clusters, a specific sample size for each cluster was allocated and participants were randomly selected. All apparently healthy adults ($\geq$ 18 y) were eligible to be recruited in our study. A total number of 2,456 adults aged 18 or older were recruited. The exclusion criteria were the presence of any systemic or dyslipidemia-related diseases according to the participants' report, chronic kidney disease, liver disease, cancer, immune system disorders and under- or over-estimation of energy intake ($<$ 800 or $>$ 4200 Kcal/day). These exclusion criteria were applied to mitigate the reverse causality. After the exclusions of around 13% of individuals, 2,033 eligible participants remained for analytical analysis. All participants signed an informed consent form. This study was approved by the ethic committee of Isfahan University of Medical Sciences in 2016 (registration number: IR.MUI.RC.1395.4.077).

Smoking and socio-demographic information including age, sex, and socioeconomic status (SES) were assessed by a self-administered questionnaire. More details about the study participants and methodology have been published elsewhere [19]. Physical activity levels were recorded using a validated Persian version of the International Physical Activity Questionnaire (IPAQ) and expressed as metabolic equivalent hours per week (MET-h / week) [20, 21].

## Dietary intake

Dietary intake, over the preceding year, was evaluated by a validated, 110-item, semi-quantitative food frequency questionnaire (FFQ) [22] and completed by trained interviewers. Each food item was examined based on a popular portion size for it. Nine possible frequency of consumption categories from never/seldom to more than 6 times/day were calculated for each individual and food item. According to the weight of each portion size and frequency of consumption, the average intake of each food item (grams/day) was estimated for the entire sample. Then, energy and nutrients intakes were calculated by means of Nutritionist IV software which was adjusted for Iranian foods.

## Assessment of plant-based dietary index

Considering the food composition of various food items, they were classified into 21 main groups. These, then, were classified into three groups namely animal sources, healthy, and unhealthy plant-based foods. The development of three plant-based diet indices (PDI, hPDI and uPDI) has also been described based on previous studies [10, 23, 24]. Healthy plant foods consisted of healthy vegetables, whole grains, fruits, nuts, legumes, vegetable oils, and tea/coffee, while unhealthy plant foods were fruit juices, potatoes, refined grains, sweets, desserts, and drinks. All foods derived from animal sources constituted animal foods. All food groups were divided into deciles (ten groups with almost identical sample size) and a score of 1-10 was assigned to each decile. Accordingly, for plant-based diet indices, scores of 10 and 1 were respectively given to individuals at the highest and the lowest deciles of all plant-based foods, regardless of their health properties. Other deciles proportionally scored between 1 and 10. Animal food groups scored conversely. In other words, participants with the highest intake received a score of 1 and those with the lowest intake score 10. For hPDI, the highest and lowest consumption of healthy plant foods scored 10 and 1, respectively, while the highest and lowest consumption of unhealthy plant foods and animal food items received a score of 1 and 10, respectively. For uPDI, participants with higher intake of unhealthy plant-based foods were given higher scores (score 10 for decile 10 and score 1 for decile 1), and healthy plant- and animal-based foods scored reversely (score 1 for decile 10 and score 10 for decile 1). Then all scores were summed to obtain the final score for each index, ranging from 21 to 210. Higher scores represent greater adherence to each score [15].

## Mental health assessment

The Symptoms of anxiety and depression were assessed using a validated Iranian version of the Hospital Anxiety and Depression Scale (HADS) [25]. This simple questionnaire includes two separate sections with 7 items with a 4-point rating scale. The final obtained scale will range from 0 (the lowest degree of anxiety and depression) to 21 (the greatest degree of anxiety or depression). A scores of $\leq 7$ in each section was considered not to have depression or anxiety symptoms (normal health status) and a score of $\geq 8$ indicated the presence of depression or anxiety symptoms[26].

## Statistical analysis

All PDI, hPDI, and uPDI were categorized into quartiles. Normality assumption was checked graphically and also by Kolmogorov-Smirnov test. General sample characteristics were expressed as percentage for categorical and mean ± standard deviation (SD) for continuous variables. The differences between quartiles were examined either applying a chi-square test for categorical variables or one-way analysis of variance (ANOVA) for continuous variables and dietary intake. We applied the Kruskal–Wallis or robust Brown–Forsyth tests to evaluate means across quartiles when the assumptions for one-way analysis of variance were not met. The mean depression and anxiety scores across quartiles of PDI, hPDI, and uPDI were compared using Kruskal–Wallis for the crude model, and using ANCOVA test in the multivariable-adjusted model. The log transformation was used for non-normal variables in ANCOVA. The crude and multivariable-adjusted odds ratio (OR) and 95% confidence intervals (CIs) for having depression or anxiety across quartiles of PDI, hPDI, uPDI were estimated applying multiple logistic regression. In the first adjusted model, the confounding effects of age, sex and energy intake were controlled. Model 2 was additionally controlled for marital status, physical activity, education level, smoking. Statistical Package for Social Sciences (SPSS) version 25 (IBM Corp, Armonk, NY, USA) was used for data analysis. $P < 0.05$ was considered statistically significant.

## Results

**Table 1** illustrates the general characteristics of participants according to quartiles of PDI, hPDI, uPDI. People with higher PDI and uPDI scores were younger than those with lower scores of PDI and uPDI. In contrast, the greater adherence to hPDI, the older participants were. Individuals in the fourth quartile of PDI were more probably to be current smoker but those in the highest quartile of hPDI were less likely to be smoker. The distribution of current smokers across the quartiles of uPDI was similar. Comparison of these variables between subjects with and without depression and subjects with and without anxiety demonstrated significant difference for all variables apart from physical activity level and smoking status (**S1 Table**).

Participants' dietary intake across quartiles of PDI, hPDI, and uPDI are shown in **Table 2**. Compared with those with lower scores of PDI, the intake of all food groups was constantly higher in subjects who had greater adherence to the PDI. The one exception to this trend was dairy products which did not significantly differ between PDI categories. In contrast, higher scores of hPDI were associated with lower energy intake and lower consumption of all three macronutrients and different fatty acids, fiber, unhealthy vegetables, nuts, refined grains, meat, fish and sea foods, dairy products, fast foods, sweet desserts and sweet drinks, while the intakes of healthy vegetables, fruit, whole grains, and legumes increased by the sores of hPDI. Likewise, greater adherence to uPDI was associated with lower intakes of most food groups, but not unhealthy vegetables, refined grains, sweet desserts and sweet drinks which were consumed in higher amounts in individuals with higher uPDI scores as compared to individuals with lower uPDI scores. Dietary intakes comparison between subjects with and without depression and subjects with and without anxiety are shown in **S2 Table**.

**Table 3** shows the mean and standard errors (SE) of psychological disorders (depression and anxiety) in crude and adjusted models across quartiles of PDI, hPDI, uPDI. No significant association was found between PDI and any of the psychological disorders either in the crude or adjusted model. In the crude model, individuals in the higher quartile of hPDI had a significantly higher mean of depression with no significant difference in anxiety score in comparison with participants in the lowest quartile. Regarding uPDI, participants in the highest quartile

**Table 1. General characteristics of participants across the quartiles of PDI, hPDI, uPDI scores.**

| | PDI | | | | | hPDI | | | | | uPDI | | | | |
|---|---|---|---|---|---|---|---|---|---|---|---|---|---|---|---|
| | $Q_1$ | $Q_2$ | $Q_3$ | $Q_4$ | P value | $Q_1$ | $Q_2$ | $Q_3$ | $Q_4$ | P value | $Q_1$ | $Q_2$ | $Q_3$ | $Q_4$ | P value |
| Age (y)[a] | 41.8 ±15.4 | 40.5 ±14.3 | 38.9 ±12.9 | 37.7 ±12.2 | <0.0001 | 34.6 ±11.6 | 38.4 ±12.7 | 40.8 ±14.0 | 45.4 ±14.8 | <0.0001 | 41.8 ±14.1 | 38.5 ±13.0 | 38.9 ±13.5 | 39.5 ±14.5 | 0.001 |
| Physical activity[b] | 334.1 ±459.8 | 408.9 ±573.9 | 491.2 ±599.6 | 577.9 ±653.9 | <0.0001 | 488.9 ±568.6 | 473.6 ±624 | 437.7 ±620.2 | 410.5 ±506.2 | 0.024 | 531.1 ±638.7 | 468.8 ±569.8 | 422.9 ±530.3 | 388.5 ±576.8 | <0.0001 |
| Male (%)[c] | 41.0 | 47.4 | 51.4 | 51.3 | 0.002 | 58.4 | 52.1 | 45.9 | 34.3 | <0.0001 | 52.3 | 49.4 | 45.3 | 44.3 | 0.036 |
| Married (n (%))[c] | 82.5 | 82.5 | 82.9 | 82.8 | 0008 | 77.1 | 84.9 | 82.9 | 86.0 | <0.0001 | 87.4 | 82.4 | 81.6 | 72.9 | 0.003 |
| Education year (n (%))[c] | | | | | 0.004 | | | | | <0.0001 | | | | | <0.0001 |
| 0-5y | 27.1 | 27.2 | 19.5 | 19.0 | | 14.1 | 22.0 | 26.4 | 30.3 | | 15.3 | 17.7 | 25.3 | 34.4 | |
| 5-127 | 45.1 | 43.0 | 48.0 | 46.6 | | 45.8 | 48.8 | 42.7 | 45.6 | | 48.1 | 46.2 | 44.7 | 44.1 | |
| >12y | 27.7 | 29.8 | 32.4 | 34.4 | | 40.0 | 29.2 | 31.0 | 24.0 | | 36.6 | 36.1 | 30.1 | 21.5 | |
| Current smoker (n (%))[c] | 6.0 | 9.7 | 13.2 | 17.3 | <0.0001 | 16.2 | 11.2 | 12.1 | 6.5 | <0.0001 | 10.6 | 12.7 | 10.8 | 12.1 | 0.66 |

PDI, overall plant-based diet index; hPDI, healthful plant-based diet index; uPDI, unhealthful plant-based diet index.

Values are mean±SD for continuous variables and percentage for dichotomous variables.

[a] p-value obtained based on robust Brown–Forsyth test.

[b] p-value obtained based on Kruskal–Wallis test.

[c] p-value obtained based on Chi-square test.

had higher levels of depression and anxiety than those in the lowest quartile. However, after adjustment for potential confounders, these associations remained no longer significant.

**Table 4** presents the odds ratio and 95% confidence intervals for depression and anxiety in crude and adjusted models across quartiles of PDI, hPDI, and uPDI. In all models, there was no significant association between PDI and depression and anxiety. Although individuals in the top quartile of hPDI tended to have higher risk of depression compared with those in the bottom quartile (OR= 1.35, 95% CI: 0.99, 1.84; P= 0.062), this tendency was eliminated in adjusted models. Despite a direct association between hPDI and anxiety in the crude model (P=0.031), no significant association was observed after adjusting for potential covariates (OR for the fourth quartile=1.16; 95% CI: 0.81, 1.66; P=0.604). A positive association was found between uPDI and depression (OR for the fourth quartile= 2.07; 95% CI: 1.49, 2.87; P<0.0001) and anxiety (OR for the fourth quartile= 1.56; 95% CI: 1.14, 2.14; P= 0.001) in the crude model which remained significant even in the fully-adjusted model (for depression, OR for the fourth quartile= 1.96; 95% CI: 1.34, 2.85; P= 0.001), and for anxiety, OR for the fourth quartile= 1.53; 95% CI: 1.07, 2.19; P=0.008) in both the crude and fully adjusted models. Stratified analysis by sex revealed similar results (**S3 Table**).

## Discussion

Our main findings showed that the PDI and hPDI were not associated with depression and anxiety, while higher scores of uPDI were associated with a higher risk of depression and anxiety. Adjustment for potential covariates did not considerably change these associations. This is the first multi-centric study amongst Iranians in this regard which can consider differences in dietary patterns derived from variations socioeconomic classes and lifestyle in various geographical regions.

**Table 2. Dietary intakes of study participants across the quartiles of PDI, hPDI, and uPDI scores.**

| | PDI | | | | P value | hPDI | | | | P value | uPDI | | | | P value |
|---|---|---|---|---|---|---|---|---|---|---|---|---|---|---|---|
| | Q₁ (N=503) | Q₂ (N=486) | Q₃ (N=559) | Q₄ (N=484) | | Q₁ (N=518) | Q₂ (N=518) | Q₃ (N=503) | Q₄ (N=493) | | Q1 (N=515) | Q₂ (N=510) | Q₃ (N=502) | Q₄ (N=506) | |
| Energy (kcal/d)[a] | 1806.28 ±573.20 | 2060.78 ±636.96 | 2416.40 ±720.36 | 2896.54 ±752.30 | <0.0001 | 2784.64 ±779.00 | 2317.77 ±740.55 | 2133.42 ±722.76 | 1920.14 ±618.85 | <0.0001 | 2669.60 ±790.7 | 2403.95 ±724.99 | 2245.34 ±725.99 | 1851.82 ±665.18 | <0.0001 |
| Carbohydrate (g/day)[a] | 214.85 ±74.69 | 255.14 ±88.07 | 298.99 ±93.19 | 362.93 ±96.11 | <0.0001 | 342.22 ±102.96 | 287.35 ±102.43 | 261.58 ±96.01 | 237.66 ±80.82 | <0.0001 | 311.39 ±98.55 | 295.68 ±94.56 | 285.66 ±100.29 | 238.30 ±106.56 | <0.0001 |
| Protein (g/day)[a] | 84.79 ±32.16 | 86.60 ±33.36 | 97.52 ±35.83 | 109.18 ±34.37 | <0.0001 | 111.42 ±34.65 | 94.83 ±33.97 | 90.23 ±33.98 | 80.87 ±31.46 | <0.0001 | 119.90 ±33.10 | 101.12 ±29.81 | 88.67 ±29.38 | 67.88 ±26.55 | <0.0001 |
| Fat (g/day)[a] | 70.87 ±29.02 | 82.34 ±30.45 | 97.84 ±37.12 | 117.00 ±38.67 | <0.0001 | 114.04 ±40.9 | 93.42 ±35.16 | 84.09 ±34.27 | 75.51 ±30.01 | <0.0001 | 108.65 ±41.30 | 96.12 ±37.29 | 88.00 ±35.89 | 79.94 ±28.73 | <0.0001 |
| SFA (g/day)[c] | 28.23 ±13.77 | 30.17 ±12.44 | 34.58 ±17.48 | 40.70 ±16.67 | <0.0001 | 40.87 ±17.16 | 33.61 ±13.44 | 31.69 ±17.60 | 27.12 ±11.83 | <0.0001 | 40.21 ±20.70 | 34.67 ±13.95 | 31.25 ±13.57 | 27.36 ±11.18 | <0.0001 |
| MUFA (g/day)[c] | 21.62 ±9.64 | 25.17 ±10.22 | 30.15 ±12.94 | 35.35 ±12.73 | <0.0001 | 35.24 ±13.46 | 28.55 ±11.25 | 25.55 ±11.38 | 22.66 ±10.44 | <0.0001 | 33.29 ±13.94 | 29.06 ±12.45 | 26.51 ±11.54 | 23.36 ±9.93 | <0.0001 |
| PUFA (g/day)[c] | 18.51 ±9.32 | 23.05 ±10.55 | 27.99 ±12.39 | 35.82 ±14.37 | <0.0001 | 32.16 ±14.49 | 26.93 ±13.07 | 23.79 ±12.61 | 22.15 ±10.82 | <0.0001 | 30.87 ±14.69 | 27.66 ±13.56 | 25.36 ±13.05 | 21.31 ±9.92 | <0.0001 |
| Fiber (g/d)[b] | 18.00 ±6.65 | 20.43 ±7.64 | 23.61 ±7.62 | 28.52 ±8.01 | <0.0001 | 24.35 ±8.10 | 22.12 ±8.51 | 22.02 ±8.71 | 21.98 ±8.17 | <0.0001 | 28.00 ±7.82 | 23.98 ±7.01 | 21.89 ±7.34 | 16.54 ±7.19 | <0.0001 |
| Healthy vegetables (g/day)[a] | 240.41 ±122.57 | 271.35 ±122.14 | 301.96 ±124.54 | 366.77 ±137.13 | <0.0001 | 288.21 ±127.02 | 283.19 ±125.53 | 297.04 ±141.41 | 311.93 ±142.87 | 0.004 | 368.19 ±138.39 | 314.34 ±126.90 | 274.96 ±119.34 | 220.07 ±105.72 | <0.0001 |
| Fruits (g/day)[c] | 206.47 ±132.27 | 225.37 ±150.00 | 253.16 ±152.95 | 305.31 ±154.19 | <0.0001 | 246.74 ±143.20 | 230.83 ±148.21 | 247.22 ±152.26 | 265.68 ±167.08 | 0.01 | 356.92 ±145.10 | 266.99 ±130.58 | 225.16 ±133.31 | 137.80 ±113.73 | <0.0001 |
| Unhealthy vegetables (g/day)[c] | 8.31 ±15.72 | 13.60 ±18.05 | 21.53 ±24.99 | 32.40 ±31.57 | <0.0001 | 30.22 ±29.79 | 19.58 ±23.84 | 16.28 ±24.02 | 9.15 ±15.26 | <0.0001 | 15.60 ±23.11 | 19.66 ±25.53 | 20.49 ±26.79 | 20.14 ±24.43 | <0.0001 |
| Legumes (g/day)[c] | 21.20 ±19.56 | 21.43 ±22.18 | 26.82 ±28.40 | 37.38 ±36.19 | <0.0001 | 24.71 ±26.49 | 26.24 ±28.48 | 26.74 ±25.42 | 29.64 ±31.35 | 0.015 | 41.00 ±35.25 | 27.25 ±26.48 | 21.65 ±23.91 | 16.38 ±16.81 | <0.0001 |
| Nuts (g/day)[c] | 14.81 ±19.54 | 20.08 ±24.79 | 27.35 ±28.71 | 39.56 ±33.34 | <0.0001 | 30.11 ±31.13 | 25.25 ±26.25 | 24.70 ±25.42 | 21.39 ±26.47 | <0.0001 | 37.64 ±33.45 | 28.07 ±30.09 | 22.44 ±25.08 | 13.23 ±16.93 | <0.0001 |
| Refined grains (g/day)[c] | 169.49 ±124.57 | 210.85 ±138.92 | 257.62 ±151.49 | 298.43 ±147.13 | <0.0001 | 323.95 ±157.37 | 245.36 ±148.74 | 202.14 ±29.40 | 161.30 ±107.68 | <0.0001 | 209.24 ±132.51 | 236.28 ±143.43 | 246.19 ±147.07 | 246.41 ±168.37 | <0.0001 |
| Whole grains (g/day)[c] | 80.73 ±90.73 | 90.47 ±100.42 | 89.12 ±99.12 | 99.57 ±99.12 | 0.001 | 55.83 ±74.50 | 85.45 ±97.79 | 97.18 ±124.29 | 122.90 ±102.75 | <0.0001 | 118.02 ±95.74 | 102.74 ±97.74 | 95.65 ±106.36 | 42.36 ±69.74 | <0.0001 |
| Meat (g/day)[c] | 79.53 ±45.08 | 71.28 ±44.87 | 74.52 ±47.13 | 68.24 ±45.39 | <0.0001 | 87.35 ±47.65 | 74.49 ±45.82 | 72.04 ±101.41 | 59.29 ±41.58 | <0.0001 | 100.09 ±46.45 | 79.48 ±44.37 | 63.78 ±41.69 | 49.89 ±33.68 | <0.0001 |
| Fish & sea food (g/day)[c] | 15.01 ±17.04 | 13.80 ±15.53 | 16.03 ±16.94 | 16.63 ±15.75 | 0.001 | 21.71 ±19.25 | 16.36 ±15.75 | 12.68 ±43.62 | 10.46 ±13.31 | <0.0001 | 22.10 ±18.95 | 16.07 ±15.52 | 13.54 ±15.52 | 9.67 ±12.23 | <0.0001 |
| Dairy (g/day)[c] | 359.88 ±243.87 | 333.23 ±231.40 | 342.82 ±221.44 | 369.57 ±246.63 | 0.071 | 400.63 ±228.77 | 361.91 ±225.74 | 347.76 ±14.18 | 291 ±223.30 | <0.0001 | 472.95 ±265.60 | 383.91 ±219.16 | 324.93 ±205.63 | 219.58 ±165.15 | <0.0001 |
| Fast food (g/day)[c] | 8.78 ±16.24 | 10.41 ±15.99 | 14.87 ±21.66 | 19.98 ±28.21 | <0.0001 | 29.23 ±29.88 | 12.64 ±15.77 | 8.27 ±16.00 | 3.23 ±7.27 | <0.0001 | 13.37 ±19.31 | 16.00 ±25.33 | 14.50 ±22.84 | 10.19 ±17.34 | 0.001 |
| Sweet dessert (g/day)[c] | 5.44 ±8.41 | 8.69 ±12.39 | 11.78 ±14.36 | 14.96 ±15.37 | <0.0001 | 16.34 ±15.68 | 11.58 ±13.53 | 7.75 ±10.91 | 4.91 ±9.58 | <0.0001 | 8.40 ±12.80 | 11.49 ±13.95 | 11.59 ±13.52 | 9.50 ±13.03 | <0.0001 |
| Sweet drink (g/day)[c] | 15.78 ±33.31 | 28.66 ±46.80 | 40.46 ±50.53 | 60.38 ±67.41 | <0.0001 | 67.71 ±63.41 | 44.11 ±57.38 | 23.41 ±10.91 | 8.06 ±18.44 | <0.0001 | 22.47 ±44.11 | 33.90 ±50.17 | 41.29 ±56.40 | 47.84 ±58.35 | <0.0001 |

PDI, overall plant-based diet index; hPDI, healthful plant-based diet index; uPDI, unhealthful plant-based diet index.

SFA: saturated fatty acid; PUFA: polyunsaturated fatty acid; MUFA; monounsaturated fatty acid.

Values are mean ± SD

[a] p-value obtained based on robust Brown–Forsyth test.

[b] p-value obtained based on ANOVA test.

[c] p-value obtained based on Kruskal–Wallis test.

**Table 3. Mean of depression and anxiety scores across quartiles of PDI, hPDI, and uPDI scores[1].**

| | PDI | | | | | hPDI | | | | | uPDI | | | | |
|---|---|---|---|---|---|---|---|---|---|---|---|---|---|---|---|
| | Q₁ (N=503) | Q₂ (N=478) | Q₃ (N=560) | Q₄ (N=485) | P value[2] | Q₁ (N=520) | Q₂ (N=518) | Q₃ (N=504) | Q₄ (N=493) | P value[2] | Q₁ (N=517) | Q₂ (N=510) | Q₃ (N=502) | Q₄ (N=506) | P value[2] |
| **Depression** | | | | | | | | | | | | | | | |
| Model I | 4.85 ±3.77 | 4.30 ±3.33 | 4.24 ±3.58 | 4.35 ±3.56 | 0.020 | 4.21 ±3.37 | 4.20 ±3.52 | 4.52 ±3.66 | 4.81 ±3.70 | 0.018 | 3.86 ±3.07 | 4.36 ±3.45 | 4.63 ±3.59 | 4.89 ±4.03 | 0.003 |
| Model II | 4.64 ±0.16 | 4.25 ±0.16 | 4.36 ±0.14 | 4.50 ±0.17 | 0.343 | 4.60 ±0.16 | 4.32 ±0.15 | 4.43 ±0.15 | 4.40 ±0.16 | 0.650 | 3.82 ±0.16 | 4.43 ±0.15 | 4.61 ±0.15 | 4.90 ±0.16 | 0.006 |
| Model III | 4.69 ±0.16 | 4.27 ±0.15 | 4.37 ±0.14 | 4.44 ±0.17 | 0.246 | 4.58 ±0.16 | 4.31 ±0.15 | 4.44 ±0.15 | 4.44 ±0.16 | 0.736 | 3.98 ±0.16 | 4.48 ±0.15 | 4.61 ±0.15 | 4.72 ±0.16 | 0.106 |
| **Anxiety** | | | | | | | | | | | | | | | |
| Model I | 4.82 ±3.09 | 4.53 ±3.64 | 4.49 ±3.81 | 4.92 ±3.78 | 0.096 | 4.43 ±3.65 | 4.63 ±3.80 | 4.68 ±3.75 | 5.02 ±3.94 | 0.120 | 4.27 ±3.33 | 4.41 ±3.72 | 5.09 ±3.91 | 4.98 ±4.11 | 0.004 |
| Model II | 4.69 ±0.17 | 4.54 ±0.17 | 4.59 ±0.15 | 4.96 ±0.18 | 0.205 | 4.65 ±0.17 | 4.75 ±0.16 | 4.64 ±0.16 | 4.72 ±0.17 | 0.966 | 4.15 ±0.16 | 4.45 ±0.16 | 5.08 ±0.16 | 5.11 ±0.17 | 0.001 |
| Model III | 4.77 ±0.17 | 4.55 ±0.16 | 4.59 ±0.15 | 4.88 ±0.18 | 0.309 | 4.63 ±0.17 | 4.73 ±0.16 | 4.65 ±0.16 | 4.77 ±0.17 | 0.923 | 4.33 ±0.16 | 4.51 ±0.16 | 5.06 ±0.16 | 4.89 ±0.17 | 0.060 |

PDI, overall plant-based diet index; hPDI, healthful plant-based diet index; uPDI, unhealthful plant-based diet index.

[1] These values are mean±SE.

[2] Derived from Kruskal-Wallis test in crude models and from ANCOVA in multivariable-adjusted models based on log transformation of depression and anxiety due to non-normality.

Model I: Crude model.

Model II: Adjusted for age, sex, and energy intake.

Model III: Additionally adjusted for marital status, education, physical activity level, and smoking.

The direct association between uPDI and mental disorders found in the present study, including depression and anxiety, are in line with previous studies [14, 15, 27]. Despite a null association between an uPDI and stress among young women [17], in an earlier study on apparently healthy, Iranian women, greater adherence to an uPDI was associated with 91% and 31% increased risk of depression and anxiety, respectively [14]. Similarly, another analysis revealed that higher scores of uPDI may increase the risk of depression nine fold, though it does not seem to be precise adequate (95% CI: 3.96, 22.07) [15].

There are several potential mechanisms to explain the effect of a plant-based diet on psychological disorders, but the precise mechanism is still unknown. Based on some findings, anxiety adversely affects food choices and contributes to consuming higher amounts of unhealthy foods and an intake of few fruits and vegetables [28]. Therefore, patients suffering from mental disorders would find it difficult to follow a healthy plant-based diet [28]. In addition, most of the unhealthy plant foods have high glycemic index (GI) and glycemic load (GL) which might influence mental health status [29–32] through their lower contents of essential nutrients for mental health [33], or their detrimental effect on gut microbiota and inflammatory processes [34, 35]. Dysbiotic microbiota may adversely influence mood [36]. Another possible reason might be related to higher inflammatory potential of an unhealthy plant-based dietary pattern [37]. Unhealthy plant-based foods such as unhealthy vegetables, refined grains, sweet dessert and drink, and low consumption of fruits, vegetables, and whole grains are associated with elevated levels of blood inflammatory markers [38]. Higher levels of inflammatory biomarkers may disrupt neurogenesis [39] and neuroplasticity [40] or the function of serotonin [41], dopamine [42] and brain-derived neurotrophic factor. In support of this mechanism, many studies reported a direct link between depression and prolonged psychological distress

**Table 4. Crude and multivariable-adjusted odds ratios and 95% CIs for anxiety and depression across quartiles of PDI, hPDI, and uPDI scores.**

| | PDI | | | | | hPDI | | | | | uPDI | | | | |
|---|---|---|---|---|---|---|---|---|---|---|---|---|---|---|---|
| | Q1 (N=503) | Q2 (N=478) | Q3 (N=560) | Q4 (N=485) | P trend | Q1 (N=520) | Q2 (N=518) | Q3 (N=504) | Q4 (N=493) | P trend | Q1 (N=517) | Q2 (N=510) | Q3 (N=502) | Q4 (N=506) | P trend |
| **Depression** | | | | | | | | | | | | | | | |
| **Model I** | 1 | 0.73 (0.53, 1.00)[d] | 0.86 (0.64, 1.16) | 0.90 (0.66, 1.22) | 0.728 | 1 | 1.07 (0.78, 1.48) | 1.09 (0.79, 1.05) | 1.35 (0.99, 1.84) | 0.062 | 1 | 1.49 (1.06, 2.09) | 1.83 (1.31, 2.55) | 2.07 (1.49, 2.87) | <0.0001 |
| **Model II** | 1 | 0.79 (0.57, 1.10) | 1.04 (0.75, 1.44) | 1.13 (0.79, 1.64) | 0.321 | 1 | 0.91 (0.65, 1.27) | 0.83 (0.58, 1.17) | 0.87 (0.61, 1.25) | 0.428 | 1 | 1.67 (1.17, 2.37) | 1.93 (1.36, 2.75) | 2.37 (1.65, 3.41) | <0.0001 |
| **Model III** | 1 | 0.75 (0.54, 1.06) | 0.99 (0.71, 1.38) | 1.03 (0.71, 1.51) | 0.615 | 1 | 0.90 (0.64, 1.27) | 0.83 (0.58, 1.18) | 0.89 (0.62, 1.29) | 0.517 | 1 | 1.69 (1.12, 2.30) | 1.80 (1.26, 2.85) | 1.96 (1.34, 2.85) | 0.001 |
| **Anxiety** | | | | | | | | | | | | | | | |
| **Model I** | 1 | 0.88 (0.65, 1.19) | 0.78 (0.58, 1.06) | 1.06 (0.78, 1.42) | 0.923 | 1 | 1.33 (0.97, 1.80) | 1.30 (0.95, 1.77) | 1.51 (1.11, 2.05) | 0.013 | 1 | 1.30 (0.94. 1.79) | 1.78 (1.31, 2.42) | 1.56 (1.14, 2.14) | 0.001 |
| **Model II** | 1 | 0.94 (0.69, 1.29) | 0.89 (0.65, 1.24) | 1.18 (0.82, 1.68) | 0.497 | 1 | 1.27 (0.91, 1.76) | 1.13 (0.8, 1.58) | 1.12 (0.79, 1.6) | 0.743 | 1 | 1.45 (1.04, 2.02) | 1.98 (1.42, 2.74) | 1.87 (1.32, 2.65) | <0.0001 |
| **Model III** | 1 | 0.91 (0.66, 1.25) | 0.86 (0.62, 1.19) | 1.09 (0.76, 1.57) | 0.777 | 1 | 1.26 (0.91, 1.76) | 1.14 (0.81. 1.62) | 1.16 (0.81, 1.66) | 0.604 | 1 | 1.38 (0.98, 1.94) | 1.82 (1.30, 2.54) | 1.53 (1.07, 2.19) | 0.008 |

PDI, overall plant-based diet index; hPDI, healthful plant-based diet index; uPDI, unhealthful plant-based diet index.

Model I: Crude model.

Model II: Adjusted for age, sex, and energy intake.

Model III: Additionally adjusted for marital status, education, physical activity level, and smoking.

and inflammatory pathways in the brain [43–45]. Lower protein intake may also adversely influence mental health status [46, 47].

In stark contrast with some previous evidence, we failed to find any significant association for either PDI or hPDI with depression or anxiety [27, 48, 49]. In a recent study among Iranian women, an inverse link was reported between hPDI and psychological disorders [27]. Consistently, a systematic review on controlled trials demonstrated that a PDI could significantly improve psychological well-being and ameliorate depressive symptoms in patients with type 2 diabetes mellitus [50]. In the GAZEL cohort, healthy dietary pattern, defined by the consumption of vegetables, was associated with fewer depressive symptoms in men and women [51]. Furthermore, in Chinese older people, the highest quartile of "vegetables-fruits" pattern score was associated with a decreased risk of incident depression compared to the lowest quartile at baseline [52]. For instance, a study has shown that total protein intake and protein intake from milk and milk products were negatively associated with depression [53]. This might be attributable to the effect of tryptophan on mood state and cognitive functions [54, 55]. These inconsistencies between studies might be owing to diversity in methodology. For instance, energy adjustment by residual method or as a confounder may potentially influence the results. However, although differences in methodology might be a reason for such discrepancies between studies, it is unlikely that methods used for energy adjustment be a determinant. Indeed, recent evidence has suggested that results obtained by the residual model or the standard method are the same [56].

Whereas healthy plant-based foods are rich sources of various antioxidants and exert anti-inflammatory properties, they could not decrease depression and anxiety risk in our study

population, which contradicts with some earlier studies [27]. The exact reason behind this is not clear but possibly it could be attributed to the overall patterns of hPDI in our study population. In fact, despite statistically significant differences between quartiles for some healthy plant-based nutrients or foods, such as fiber, healthy vegetables, fruit and legumes, the variations in overall intake were fewer than one serving between the highest and the lowest intakes for all which do not allow them to exert their beneficial effects on health status. For fiber, this difference was approximately two grams, while in comparison with studies which found significant associations, differences were much larger between different categories [27]. In other words, the homogeneity in dietary intakes might explain the null association found between hPDI and depression and anxiety in our study. Besides that, differences in study population, like higher mean age of our participants and examining both sexes compared with Zamani et al.'s [14] might be other possible explanations for our findings. For instance, sex-differences in using coping strategies [46] or varying risk factors for depression in different age categories [57] may impact the PDI-mental health association.

Our study has several strengths. First, our study population is a large multicentric sample of Iranian adults and therefore our results have a significant external validity to extrapolate our findings to the Iranian. Second, since this study was conducted amongst healthy subjects, the confounding effect of diseases on mental health status is eliminated. Third, using a validated FFQ provides us with assessing habitual consumption of most of the food items over a long run and consequently a more precise classification of food groups. Forth, given that various plant foods may exert different health consequences, for instance refined and whole grains, in this study, we categorized them into two healthy and unhealthy groups and examined their impacts and patterns separately in our study population. Fifth, all questionnaires were completed by trained interviewers which provides more reliable and precise responses.

There are several limitations of the present study that should be acknowledged. The cross-sectional design of this study does not allow us to draw causal inferences. For instance, anxiety may cause people to have greater tendency towards unhealthy food choices. Although all questionnaires were completed by interviewers, it is possible that respondents have answered questions in a manner that they seem psychologically healthy. Moreover, in the present study, only demographic and lifestyle variables were examined and the confounding effect of unmeasured and residual confounders, such as food preferences and adhering to any specific type of diet, cannot be completely ruled out. Finally, the FFQ is a memory-reliable tool and subject to both random and systematic measurement errors, leading to misclassification of dietary intakes, which is an inevitable bias in nutritional epidemiological research. More reliable associations might have been acquired provided that depression, anxiety and dietary intakes were measured by the means of more sophisticated tools, such as physician diagnosis or a combination of instruments (e.g. a 24h recall beside FFQ [58]).

In conclusion, our results suggest that the greater adherence to an unhealthy plant-based diet, the higher risk of depression and anxiety. However, neither PDI nor hPDI was pertinent to depression and anxiety. Plant-based diets' composition is similar to a provegetarian diet, identified by Martínez-González et al. [59]. Long-term adherence to a pure vegetarian diet is not convenient for many individuals, however, consuming a greater proportion of daily energy intake from plant-derived foods is an easier goal which can be achieved by many individuals. In addition, due to great heterogeneity in terms of the association of vegetarian diets with mental health status [8], our results can shed light on diets, mainly based on plant foods, and mental health relations. Accordingly, though an uPDI in our study population is a risk factor for depression and anxiety, overall PDI or hPDI cannot be a protective factor against them. However, there is still few data in this regard and our results need to be confirmed by large prospective cohort studies.

## Supporting information

**S1 Table. General characteristics of participants across the quartiles of PDI, hPDI, uPDI scores, stratified by depression and anxiety status.**
(DOCX)

**S2 Table. Dietary intakes of study participants across the quartiles of PDI, hPDI, uPDI scores, stratified by depression and anxiety status.**
(DOCX)

**S3 Table. Crude and multivariable-adjusted odds ratios and 95% CIs for anxiety and depression across quartiles of PDI, hPDI, and uPDI scores stratified by sex.**
(DOCX)

## Acknowledgments

We greatly appreciate the help from all staff in the five studied counties with their assistance in data collection and conducting intervention activities. All authors declare no other competing interests.

## Author Contributions

**Conceptualization:** Fahimeh Haghighatdoost, Noushin Mohammadifard, Nizal Sarrafzadegan.

**Data curation:** Noushin Mohammadifard.

**Formal analysis:** Razieh Hassannejad.

**Funding acquisition:** Nizal Sarrafzadegan.

**Investigation:** Noushin Mohammadifard, Farid Najafi, Hossein Farshidi, Masoud Lotfizadeh, Tooba Kazemi, Nizal Sarrafzadegan.

**Methodology:** Noushin Mohammadifard, Hamidreza Roohafza, Erika Aparecida Silveira, Nizal Sarrafzadegan.

**Project administration:** Nizal Sarrafzadegan.

**Resources:** Farid Najafi, Hossein Farshidi, Masoud Lotfizadeh, Tooba Kazemi, Nizal Sarrafzadegan.

**Software:** Noushin Mohammadifard, Simin Karimi.

**Supervision:** Nizal Sarrafzadegan.

**Validation:** Noushin Mohammadifard.

**Writing – original draft:** Fahimeh Haghighatdoost, Atena Mahdavi, Cesar de Oliveira.

**Writing – review & editing:** Fahimeh Haghighatdoost, Atena Mahdavi, Cesar de Oliveira.

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
