## [Decision Letter · Decision Letter 0]

28 Nov 2022

PONE-D-22-15117The relationship between a plant-based diet and mental health: evidence from a cross-sectional multicentric community trial (LIPOKAP Study)PLOS ONE

Dear Dr. Mohammadifard,

Thank you for submitting your manuscript to PLOS ONE. After careful consideration, we feel that it has merit but does not fully meet PLOS ONE’s publication criteria as it currently stands. Therefore, we invite you to submit a revised version of the manuscript that addresses the points raised during the review process.

We look forward to receiving your revised manuscript.

Kind regards,

Mohammad Hossein Ebrahimi

Academic Editor

PLOS ONE

Journal Requirements:

This study has been funded by Pfizer company with grant number of 11531879. Dr Cesar de Oliveira is supported by the Economic and Social Research Council (ESRC) (grant ES/T008822/11).

This study received funding from Pfizer company (#11531879). The funder was not involved in the study design, collection, analysis, interpretation of data, the writing of this article or the decision to submit it for publication. 

However, funding information should not appear in the Acknowledgments section or other areas of your manuscript. We will only publish funding information present in the Funding Statement section of the online submission form. 

This study has been funded by Pfizer company with grant number of 11531879. Dr Cesar de Oliveira is supported by the Economic and Social Research Council (ESRC) (grant ES/T008822/11).

NO authors have competing interests

Reviewers' comments:

Reviewer's Responses to Questions

**Comments to the Author**

1. Is the manuscript technically sound, and do the data support the conclusions?

Reviewer #1: Partly

Reviewer #2: Partly

Reviewer #3: Partly

Reviewer #4: Yes

Reviewer #5: Partly

Reviewer #6: Partly

2. Has the statistical analysis been performed appropriately and rigorously? 

Reviewer #1: No

Reviewer #2: No

Reviewer #3: Yes

Reviewer #4: Yes

Reviewer #5: I Don't Know

Reviewer #6: Yes

3. Have the authors made all data underlying the findings in their manuscript fully available?

Reviewer #1: No

Reviewer #2: Yes

Reviewer #3: No

Reviewer #4: Yes

Reviewer #5: No

Reviewer #6: No

4. Is the manuscript presented in an intelligible fashion and written in standard English?

Reviewer #1: No

Reviewer #2: Yes

Reviewer #3: No

Reviewer #4: Yes

Reviewer #5: Yes

Reviewer #6: Yes

5. Review Comments to the Author

Reviewer #1: The manuscript describes the analysis of HADS and a semi-quantitative food frequency questionnaire in 2033 subjects from Iran. Several key aspects need a major revision (coherence between title, main aim and tools used: use of validated instruments; details of sample sizes for the different groups; adequate exploration of confounding variables; details of statistical analyses carried out;  presentation of data, adequate discussion of hypotheses and findings; among others).

Reviewer #2: Authors based on a relatively large cross-sectional analysis investigated the association of a priori food pattern i.e., PDI and its components with two mental disorders including anxiety and depression in Iran

The following points are suggested

Novelty of study should be declared.

The first sentence in methods section should be revised.

More details about the study population and sampling methods and generally about main study should be presented.

More details about the validity and reliability of all instruments used in this study should be presented.

More important: PDI as a priori pattern should be constructed in energy adjusted format then use it in association analysis

The following sentence is not correct “The mean depression and anxiety scores across quartiles of PDI, hPDI, and uPDI were 8 169 compared in crude and multivariable-adjusted models by using Kruskal–Wallis and ANCOVA, 170 respectively.”

Some important statistical analysis revision are required and should be presented revised analysis in results section.

For selecting appropriate confounders it is needed to compare variable you have presented in table 1 and macro and micro nutrients between depressed and non-depressed and anxious and non-anxious , you should have table for such comparisons and select relevant confounders based on these comparisons and those you compared across quartiles of PDIs and redo the analyses.

Due to gender difference based on both PDI scores and mental disorders subgroup analysis based on gender is necessary.

Reviewer #3: Materials and Methods section need to be more elaborated with the clarification of the following issues:

1. It is not clear how the study divided food items into 21 main groups, and subsequently into three groups-animal sources, healthy, and unhealthy plant-based foods.

2. Validity and reliability of the food frequency questionnaire in local language is needed.

3. Was there any statistical basis for the sample size calculation? If so, then it should be mentioned.

Reviewer #4: Summary of the research and overall impression

The manuscript presents a detailed study in a relatively new area of research- exploring the relationship between plant- based diets and mental health. The promotion, prevention and treatment of mental health is recognized in the SDGs. The manuscript is technically sound, written in an intelligible fashion and in standard English. The statistical analysis has been performed appropriately and the data supports the conclusions. However, the minor areas of improvement raised below should be addressed to enhance the quality.

Minor areas of Improvement

Methods

Mental Health Assessment- Line 158- please explain what informed the cut off of f ≤7

Discussion

The sentence beginning on line 255 may need to be reviewed for it to be clearer.

Discuss in simple and clear terms the implication of the findings to the practice of public health

The manuscript would benefit from a spell-check and minor English copy-editing

Reviewer #5: Kindly get it reviewed by a colleague who is not a co-author so that grammatical mistakes and punctuation could be corrected. Also refer to the reviewer comments in the attached pdf file and address those.

Reviewer #6: 1. The authors state that nearly 13.1% of the individuals were excluded from the parent study (LIPOKAP). However, the has not been dealt explicitly in the discussion.

2. What do the authors mean by semi-quantitative food frequency questionnaire? Was a different FFQ used? If yes, kindly include in supplement.

3. Basis of categorisation of PDI into animal sources, healthy and unhealthy has not been explicitly explained. The reference no. 13 also does not clarify the doubt

4. Was the normality of data checked before using Krushkal Wallis test?

5. Authors have indicated in the introduction that existing literature lacks evidence on temporal association of PDI and depression / anxiety. However, in the discussion authors have tried to establish temporal association which cannot be done in a cross sectional type of study design

6. Overanalyses on the part of authors dealing on the beneficial effect of PDI can be noted. stating that the study population was healthy seems to be over exaggerated in the discussion.

6. PLOS authors have the option to publish the peer review history of their article (what does this mean?). If published, this will include your full peer review and any attached files.

Reviewer #1: No

Reviewer #2: No

Reviewer #3: No

Reviewer #4: No

Reviewer #5: No

Reviewer #6: No

---

## [Author Response · Author response to Decision Letter 0]

25 Feb 2023

Editor's and Reviewers' comments:

Reviewer #1: 

The manuscript describes the analysis of HADS and a semi-quantitative food frequency questionnaire in 2033 subjects from Iran. Several key aspects need a major revision (coherence between title, main aim and tools used: use of validated instruments; details of sample sizes for the different groups; adequate exploration of confounding variables; details of statistical analyses carried out; presentation of data, adequate discussion of hypotheses and findings; among others).

Response: Thanks for your comment. We tried to improve the above-mentioned concerns through the manuscript according to the other reviewers’ comments. However, since the reviewer 1’s concerns were not exactly determined, we do not know whether our modifications can meet the reviewer’s concerns. We would be so grateful if the reviewer exactly explains each concern needs what modification or correction.

Reviewer #2: 

Authors based on a relatively large cross-sectional analysis investigated the association of a priori food pattern i.e., PDI and its components with two mental disorders including anxiety and depression in Iran.

The following points are suggested

1-Comment: Novelty of study should be declared.

Agreed. We highlighted the differences of our work with earlier ones in the third paragraph of discussion (page 4, lines 94-109). We emphasized that differences in populations and their lifestyles, in the definition of plant-based diets and the main contributors to plant-based diet may affect the association. In addition, earlier studies have limited external validity since they have been conducted on small and specific populations. Therefore, our study get priority in comparison with them.

2- Comment: The first sentence in methods section should be revised.

Thanks. It has been corrected (page 5, lines 112-113).

3- Comment: More details about the study population and sampling methods and generally about main study should be presented.

Agreed. More details were provided on pages 5-6, lines 113-122 and 125-131.

4- Comment: More details about the validity and reliability of all instruments used in this study should be presented.

Agreed. Related references for instruments validity (IPAQ, SES, and dietary intakes) were added in the revised version.

5- Comment: More important: PDI as a priori pattern should be constructed in energy adjusted format then use it in association analysis.

Thanks for your comment. According to a recent published article (PMID# 8116608; Ref# 55) in this regard “Studies that seek to estimate the causal effect of ≥1 dietary components on ≥1 outcomes should clearly state their target estimands of interest and justify an adjustment strategy for estimating this effect.”. Indeed, there is no recommended method and it is subjective for the investigators. On the other hand, we did not apply residual method for the following reasons: 1) we did not reach the estimated figures by the earlier published articles on PBD and mental disorders (which were assumed to be our target), 2) obtaining the same results by the standard model (energy adjusted as a confounder) and residual method, discussed in PMID# 8116608. This matter was highlighted in the revised version (pages 12-13, lines 292-296). 

6- Comment: The following sentence is not correct “The mean depression and anxiety scores across quartiles of PDI, hPDI, and uPDI were compared in crude and multivariable-adjusted models by using Kruskal–Wallis and ANCOVA, respectively”

Thanks, this sentence is modified in Statistical analysis section (page 8, lines 190-192).

7- Comment: Some important statistical analysis revision are required and should be presented revised analysis in results section.

Thanks for your comment. According to the following comments by the reviewer, some analyses were repeated and reported in the revised version.

8- Comment: For selecting appropriate confounders it is needed to compare variable you have presented in table 1 and macro and micro nutrients between depressed and non-depressed and anxious and non-anxious , you should have table for such comparisons and select relevant confounders based on these comparisons and those you compared across quartiles of PDIs and redo the analyses.

Thank you. The main objective of this study is assessing the relationship between plant based diets and depression and anxiety. Since adherence to a specific diet affects macro- and micro-nutrients intake and imposes its effect through its different nutrients content, adjustment for either food groups or nutrients may cause over-adjustment and cannot reveal the true association. Therefore, in the present study, we just adjusted for demographic and lifestyle risk factors according to table 1, but compare dietary intakes and demographic and lifestyle risk factors between depressed and non-depressed and anxious and non-anxious. These results were reported as supplementary file (page 9, lines 207-210; page 10, lines 222-223).

9- Comment: Due to gender difference based on both PDI scores and mental disorders subgroup analysis based on gender is necessary.

Agreed. Logistic regression analysis stratified by sex was performed and its results were stated in supplementary files and results section (page 11, lines 244-245).

Reviewer #3:

Materials and Methods section need to be more elaborated with the clarification of the following issues:

1- Comment: It is not clear how the study divided food items into 21 main groups, and subsequently into three groups-animal sources, healthy, and unhealthy plant-based foods.

Thanks for your comment. It was in accordance with the original study which developed this score, and the relevant references were cited for this categorization (page 7, lines 154-162, references 10, 23, 24).

2- Comment: Validity and reliability of the food frequency questionnaire in local language is needed.

Agreed. The relevant reference was cited (page 7, line 147, reference 22).

3- Comment: Was there any statistical basis for the sample size calculation? If so, then it should be mentioned.

Agreed. More details regarding the sample size and sampling method were provided in the revised version (pages 5-6, lines 117-122).

Sample size was calculated using the following formula considering the error of type I=5%, and study power=90%. Standard error for participants’ knowledge regarding dyslipidemia was calculated using the variation range (100-0)/6= 16.7 and d (absolute margin of error) was considered to be= 1.2%, 

n=((z_(1-α/2) +z_(1-β) )^2 s^2 )/d^2 

Estimated sample size was equal to 2000 subjects.

Reviewer #4: 

Summary of the research and overall impression

The manuscript presents a detailed study in a relatively new area of research- exploring the relationship between plant- based diets and mental health. The promotion, prevention and treatment of mental health is recognized in the SDGs. The manuscript is technically sound, written in an intelligible fashion and in standard English. The statistical analysis has been performed appropriately and the data supports the conclusions. However, the minor areas of improvement raised below should be addressed to enhance the quality.

Thank you for the positive response.

Minor areas of Improvement

Methods

1- Comment: Mental Health Assessment- Line 158- please explain what informed the cut off of f ≤7

Thanks. We declared that values ≤7 shows normal health status (page 8, line 180).

Discussion

2- Comment: The sentence beginning on line 255 may need to be reviewed for it to be clearer.

Thanks, this sentence is modified (page 12, lines 282-284).

Discuss in simple and clear terms the implication of the findings to the practice of public health

Thanks very much.

The manuscript would benefit from a spell-check and minor English copy-editing.

Thanks very much. Cesar de Oliveira, one of the authors who is a native English speaker edited our manuscript for grammatical mistakes.

Reviewer #5: 

Kindly get it reviewed by a colleague who is not a co-author so that grammatical mistakes and punctuation could be corrected. Also refer to the reviewer comments in the attached pdf file and address those.

Thanks very much.

-Comment 1: actually, our study was performed in the framework of the LIPOKAP which is a community-based clinical trial. In this study, we used the baseline data of the LIPOKAP, and this point has been addressed on page 5, lines 113-116.

-Comment 2: Due to word limitation for abstract, we need to use abbreviations. However, we tried to avoid them in conclusion and objective according to the reviewer’s comment.

-Comment 3: Thanks very much. Objective was paraphrased to make it clearer for readers (page 5, lines 107-109).

-Comment 4: Both objectives of the study were combined in one sentence (page 5, lines 107-109).

-Comment 5: Given that we could not find any similar work, we estimated SD based on its calculated sample size. Details regarding sample size estimation were provided in the revised version (pages 5-6, lines 117-122).

-Comment 6: “resources” was replaced with “sources”. (page 7, line 160)

-Comment 7: We explained that “deciles (ten groups with almost identical sample size)”. (page 7, lines 161-162)

-Comment 8: “was” was replaced with “were”. (page 8, line 175)

-Comment 9: references were corrected according to the journal’s format.

Reviewer #6: 

1- Comment: The authors state that nearly 13.1% of the individuals were excluded from the parent study (LIPOKAP). However, the has not been dealt explicitly in the discussion.

Thanks for your comment. Since these subjects did not have complete information, we excluded them from analysis. Therefore, there are no data for comparing them between included and excluded subjects. Indeed, this issue does not matter for cross-sectional studies and it is a matter for cohort studies which should compare data between followed and missed to follow.

2- Comment: What do the authors mean by semi-quantitative food frequency questionnaire? Was a different FFQ used? If yes, kindly include in supplement.

Not actually. It is the same and a common expression for FFQ which can be seen in many articles. Indeed, since it does not directly ask about grams, but portions, it is called “semi-quantitative”.

3- Comment: Basis of categorisation of PDI into animal sources, healthy and unhealthy has not been explicitly explained. The reference no. 13 also does not clarify the doubt.

Thanks for your comment. Table 1 in reference 13 clearly has stated food groups and the way of scoring for each one. We just copied and pasted it here.

Table 1

Examples of food items constituting the 18 food groups (from the 1984 NHS FFQ)

 PDI hPDI uPDI

Plant Food Groups 

Healthy 

Whole grains Whole grain breakfast cereal, other cooked breakfast cereal, cooked oatmeal, dark bread, brown rice, other grains, bran, wheat germ, popcorn Positive scores Positive scores Reverse scores

Fruits Raisins or grapes, prunes, bananas, cantaloupe, watermelon, fresh apples or pears, oranges, grapefruit, strawberries, blueberries, peaches or apricots or plums Positive scores Positive scores Reverse scores

Vegetables Tomatoes, tomato juice, tomato sauce, broccoli, cabbage, cauliflower, Brussels sprouts, carrots, mixed vegetables, yellow or winter squash, eggplant or zucchini, yams or sweet potatoes, spinach cooked, spinach raw, kale or mustard or chard greens, iceberg or head lettuce, romaine or leaf lettuce, celery, mushrooms, beets, alfalfa sprouts, garlic, corn Positive scores Positive scores Reverse scores

Nuts Nuts, peanut butter Positive scores Positive scores Reverse scores

Legumes String beans, tofu or soybeans, beans or lentils, peas or lima beans Positive scores Positive scores Reverse scores

Vegetable oils Oil-based salad dressing, vegetable oil used for cooking Positive scores Positive scores Reverse scores

Tea & Coffee Tea, coffee, decaffeinated coffee Positive scores Positive scores Reverse scores

Less healthy 

Fruit juices Apple cider (non-alcoholic) or juice, orange juice, grapefruit juice, other fruit juice Positive scores Reverse scores Positive scores

Refined grains Refined grain breakfast cereal, white bread, English muffins or bagels or rolls, muffins or biscuits, white rice, pancakes or waffles, crackers, pasta Positive scores Reverse scores Positive scores

Potatoes French fries, baked or mashed potatoes, potato or corn chips Positive scores Reverse scores Positive scores

Sugar sweetened beverages Colas with caffeine & sugar, colas without caffeine but with sugar, other carbonated beverages with sugar, non-carbonated fruit drinks with sugar Positive scores Reverse scores Positive scores

Sweets and Desserts Chocolates, candy bars, candy without chocolate, cookies (home-baked & ready-made), brownies, doughnuts, cake (home-baked & ready-made), sweet roll (home-baked & ready-made), pie (home-baked & ready-made), jams or jellies or preserves or syrup or honey Positive scores Reverse scores Positive scores

Animal Food Groups 

Animal fat Butter added to food, butter or lard used for cooking Reverse scores Reverse scores Reverse scores

Dairy Skim low fat milk, whole milk, cream, sour cream, sherbet, ice cream, yogurt, cottage or ricotta cheese, cream cheese, other cheese Reverse scores Reverse scores Reverse scores

Egg Eggs Reverse scores Reverse scores Reverse scores

Fish or Seafood Canned tuna, dark meat fish, other fish, shrimp or lobster or scallops Reverse scores Reverse scores Reverse scores

Meat Chicken or turkey with skin, chicken or turkey without skin, bacon, hot dogs, processed meats, liver, hamburger, beef or pork or lamb mixed dish, beef or pork or lamb main dish Reverse scores Reverse scores Reverse scores

Misc. animal-based foods Pizza, chowder or cream soup, mayonnaise or other creamy salad dressing Reverse scores Reverse scores Reverse scores

Abbreviations: hPDI, Healthful Plant-based Diet Index; PDI, Overall Plant-based Diet Index; uPDI, Unhealthful Plant-based Diet Index

4- Comment: Was the normality of data checked before using Krushkal Wallis test?

Yes. Normality assumption was checked graphically and also by Kolmogorov-Smirnov test. This point was stated in the revised version (page 8, lines 183-184).

5- Comment: Authors have indicated in the introduction that existing literature lacks evidence on temporal association of PDI and depression / anxiety. However, in the discussion authors have tried to establish temporal association which cannot be done in a cross sectional type of study design

We agree with the reviewer that cross-sectional studies cannot illustrate temporal associations and we had highlighted this point in our limitations in the original version (page 14, lines 323-325). But we could not notice that where the reviewer means exactly. We have written our discussion with this limitation and we would be so grateful if the reviewer could tell us where they exactly mean.

6- Comment: Overanalyzes on the part of authors dealing on the beneficial effect of PDI can be noted. Stating that the study population was healthy seems to be over exaggerated in the discussion.

The aim of stating this point is highlighting the differences between earlier studies and ours. Given that mental disorders are more frequent amongst patients such as diabetic subjects, the high prevalence of outcome in the study population may lead to overestimation of the association. This matter was added in the text (page 4, lines 95-96).

---

## [Decision Letter · Decision Letter 1]

3 Apr 2023

The relationship between a plant-based diet and mental health: evidence from a cross-sectional multicentric community trial (LIPOKAP Study)

PONE-D-22-15117R1

Dear Dr. Mohammadifard,

We’re pleased to inform you that your manuscript has been judged scientifically suitable for publication and will be formally accepted for publication once it meets all outstanding technical requirements.

Kind regards,

Mohammad Hossein Ebrahimi

Academic Editor

PLOS ONE

Additional Editor Comments (optional):

Reviewers' comments:

Reviewer's Responses to Questions

**Comments to the Author**

1. If the authors have adequately addressed your comments raised in a previous round of review and you feel that this manuscript is now acceptable for publication, you may indicate that here to bypass the “Comments to the Author” section, enter your conflict of interest statement in the “Confidential to Editor” section, and submit your "Accept" recommendation.

Reviewer #4: All comments have been addressed

Reviewer #5: All comments have been addressed

Reviewer #6: All comments have been addressed

2. Is the manuscript technically sound, and do the data support the conclusions?

Reviewer #4: Yes

Reviewer #5: Yes

Reviewer #6: Yes

3. Has the statistical analysis been performed appropriately and rigorously? 

Reviewer #4: Yes

Reviewer #5: Yes

Reviewer #6: Yes

4. Have the authors made all data underlying the findings in their manuscript fully available?

Reviewer #4: Yes

Reviewer #5: Yes

Reviewer #6: Yes

5. Is the manuscript presented in an intelligible fashion and written in standard English?

Reviewer #4: Yes

Reviewer #5: Yes

Reviewer #6: Yes

6. Review Comments to the Author

Reviewer #4: (No Response)

Reviewer #5: 1-If possible, provide references to the values of appropriate parameters used or at least to the formula used in sample size calculation.

2.The relationship between a plant-based diet and mental health: evidence from a cross-sectional multicentric community trial . IF CROSS-ECTIONAL is deleted then confusion about the design will be addressed. As your multicentric study is not cross-sectional. Also remove (LIPOKAP Study) from the title but do mention it in your Methodology. Your title may look like RELATIONSHIP BETWEEN A PLANT-BASED DIET AND MENTAL HEALTH-EVIDENCE FROM A MULTICENTRIC COMMUNITY TRIAL.

Reviewer #6: The manuscript in its current form is suitable for publication. I thank the authors for addressing all the comments in an intelligent fashion by the peer reviewers.

7. PLOS authors have the option to publish the peer review history of their article (what does this mean?). If published, this will include your full peer review and any attached files.

Reviewer #4: No

Reviewer #5: No

Reviewer #6: **Yes: **Aftab Ahmad

---

## [Editor Report · Acceptance letter]

9 May 2023

PONE-D-22-15117R1 

The relationship between a plant-based diet and mental health: evidence from a cross-sectional multicentric community trial (LIPOKAP Study) 

Dear Dr. Mohammadifard:

I'm pleased to inform you that your manuscript has been deemed suitable for publication in PLOS ONE. Congratulations! Your manuscript is now with our production department. 

Kind regards, 

on behalf of

Dr. Mohammad Hossein Ebrahimi 

Academic Editor

PLOS ONE